# Diversity and inclusion: A hidden additional benefit of Open Data

**Marie-Laure Charpignon**[1,2]*, Leo Anthony Celi[3,4,5], Marisa Cobanaj[6], Rene Eber[7], Amelia Fiske[8], Jack Gallifant[3,9], Chenyu Li[10], Gurucharan Lingamallu[11], Anton Petushkov[12], Robin Pierce[13]

1 Institute for Data, Systems, and Society, Massachusetts Institute of Technology, Cambridge, Massachusetts, United States of America, 2 Broad Institute of MIT and Harvard, Cambridge, Massachusetts, United States of America, 3 Laboratory for Computational Physiology, Massachusetts Institute of Technology, Cambridge, Massachusetts, United States of America, 4 Division of Pulmonary, Critical Care and Sleep Medicine, Beth Israel Deaconess Medical Center, Boston, Massachusetts, United States of America, 5 Department of Biostatistics, Harvard T.H. Chan School of Public Health, Boston, Massachusetts, United States of America, 6 National Center for Radiation Research in Oncology, OncoRay, Helmholtz-Zentrum Dresden-Rossendorf, Dresden, Germany, 7 Montpellier Research in Management, Montpellier University, France, 8 Institute of History and Ethics in Medicine, Department of Clinical Medicine, TUM School of Medicine and Health, Technical University of Munich, Munich, Germany, 9 Department of Critical Care, Guy's and St Thomas' NHS Trust, London, United Kingdom, 10 University of Pittsburgh School of Medicine Department of Biomedical Informatics, 5607 Baum Blvd, Pittsburgh, Pennsylvania, United States of America, 11 University of Washington, Seattle, Washington, United States of America, 12 University of Michigan, Ann Arbor, Michigan, United States of America, 13 University of Exeter, Exeter, United Kingdom

* mcharpig@mit.edu

## Abstract

The recent imperative by the National Institutes of Health to share scientific data publicly underscores a significant shift in academic research. Effective as of January 2023, it emphasizes that transparency in data collection and dedicated efforts towards data sharing are prerequisites for translational research, from the lab to the bedside. Given the role of data access in mitigating potential bias in clinical models, we hypothesize that researchers who leverage open-access datasets rather than privately-owned ones are more diverse. In this brief report, we proposed to test this hypothesis in the transdisciplinary and expanding field of artificial intelligence (AI) for critical care. Specifically, we compared the diversity among authors of publications leveraging open datasets, such as the commonly used MIMIC and eICU databases, with that among authors of publications relying exclusively on private datasets, unavailable to other research investigators (e.g., electronic health records from ICU patients accessible only to Mayo Clinic analysts). To measure the extent of author diversity, we characterized gender balance as well as the presence of researchers from low- and middle-income countries (LMIC) and minority-serving institutions (MSI) located in the United States (US). Our comparative analysis revealed a greater contribution of authors from LMICs and MSIs among researchers leveraging open critical care datasets (treatment group) than among those relying exclusively on private data resources (control group). The participation of women was similar between the two groups, albeit slightly larger in the former. Notably, although over 70% of all articles included at least one author inferred to be a woman, less than 25% had a woman as a first or last author. Importantly, we found that the proportion of authors from LMICs was substantially higher in the treatment than in the



**Data Availability Statement:** Our scripts and datasets are publicly available on GitHub: https://github.com/anpetushkov/OpenVsPrivateDatasets.

**Funding:** LAC is funded by the National Institute of Health through R01 EB017205, DS-I Africa U54 TW012043-01, and Bridge2AI OT2OD032701, and by the National Science Foundation through ITEST #2148451. JG is funded by the National Institute of Health through R01 EB017205, DS-I Africa U54 TW012043-01, and Bridge2AI OT2OD032701. MLC is funded by the Eric and Wendy Schmidt Center at the Broad Institute of MIT and Harvard. The funders had no role in study design, data collection and analysis, decision to publish, or preparation of the manuscript.

**Competing interests:** The authors have declared that no competing interests exist.

control group (10.1% vs. 6.2%, p<0.001), including as first and last authors. Moreover, we found that the proportion of US-based authors affiliated with a MSI was 1.5 times higher among articles in the treatment than in the control group, suggesting that open data resources attract a larger pool of participants from minority groups (8.6% vs. 5.6%, p<0.001). Thus, our study highlights the valuable contribution of the Open Data strategy to underrepresented groups, while also quantifying persisting gender gaps in academic and clinical research at the intersection of computer science and healthcare. In doing so, we hope our work points to the importance of extending open data practices in deliberate and systematic ways.

## Author summary

In light of the significance of data access to the mitigation of bias in clinical models, we hypothesize that researchers who leverage existing open-access datasets rather than privately-owned ones are more diverse. In this brief report, we propose to test this hypothesis in the transdisciplinary and expanding field of artificial intelligence for critical care. Specifically, we compare the diversity among authors of publications leveraging open datasets, such as the commonly used MIMIC and eICU databases, with that among authors of publications relying exclusively on private datasets, unavailable to other research investigators. To measure the extent of author diversity, we characterize gender balance, geographic diversity (i.e., the number of countries with which authors are affiliated and the income categories these countries map to), and the presence of researchers from minority-serving institutions located in the United States. Furthermore, we comment on the challenges of increasing the participation of researchers from underrepresented groups and suggest changes that can be made to the current Open Data strategy to enhance representation in authorship in the next decade. By evaluating the association between data accessibility and author diversity, our study pinpoints actionable steps that the broader field of clinical AI can take to foster inclusion in the scientific community and mitigate blind spots in data preparation and/or model development.

## Introduction

The rapidly expanding field of health data science integrates two established disciplines: computer science and healthcare. It promises to address the growing complexity of healthcare systems arising from (a) the multiplicity of care delivery settings (e.g., hospital, home), (b) the increasing number of data sources–both traditional (e.g., UK BioBank, NIH All Of Us) and non-traditional (e.g., marker trajectories from wearables, social media traces of health-related behavior), and (c) their multi-modality (e.g., structured electronic health records (EHR), medical images, genome sequencing, clinical notes, voice recordings). In parallel with this ambitious endeavor, the emergence of health data science has increased the need for significant changes in the education of health professionals. Recent examples include courses in machine learning in healthcare and opportunities to shadow a team deploying clinical algorithms in hospitals. Such theoretical and practical trainings are foundational as data science plays an increasing role in the provision of healthcare. These combined skills are needed to retrospectively derive novel insights using statistical inference (e.g., estimating treatment effects using observational data), to build interpretable clinical models (e.g., predicting in-hospital mortality

in a given time horizon), and to support their prospective implementation (e.g., conducting risk analyses and identifying potential errors that can be systematically addressed). Depending on the professional role, skills in one or more of these areas are becoming increasingly necessary to apprehend real-world data used in clinical models.

## Bias in clinical models can emanate from multiple sources

The data underlying clinical models may contain biases that can be unknowingly propagated to downstream inference and prediction tasks [1,2]. Such biases can emanate from multiple sources–ranging from differences in how physicians report information in EHR and clinical notes to artifacts in images to the miscalibration of medical devices–but they can also reflect existing social determinants of health. In other words, biases encountered in clinical data are often intersectional in nature, i.e., both social and technological [3]. For example, skin tone affects the accuracy of pulse oximetry but is rarely considered in trials measuring medical device performance [4,5,6,7]. Interrogating existing datasets and fully understanding the underlying biases requires a multidisciplinary examination involving more than a single data analyst or research group. Instead, the cooperation of clinicians, engineers, data scientists, social scientists, and industry partners is greatly needed. Indeed, studies have shown that research groups with more diverse expertise are more effective in identifying or addressing issues of bias [8].

## Study hypothesis and contribution

In this study, we seek to understand the role of open data and diversity in research expertise towards mitigating biases affecting clinical models. We hypothesize that the groups of researchers who leverage existing open-access datasets are more diverse than those using privately-owned datasets. In what follows, we explain our rationale and motivation for analyzing the profile of researchers who use open vs. private datasets.

## Many existing approaches to mitigate bias occur downstream of model development

The timing of efforts to address bias may be critical. When, in the lifecycle of a clinical model, are interventions to mitigate bias most effective? Healthcare systems can deploy interventions either downstream or upstream of the model development phase to mitigate the repercussions of data biases. Researchers in the field of ethics in artificial intelligence (AI) for health have participated in this effort by exploring several downstream approaches. Notably, biases can be mitigated after model fitting and/or deployment [9] through the use of explainable AI (XAI) tools [10,11] that identify biased features contributing to discriminatory outcomes (e.g., by decomposing individual predicted risk scores).

## The public release of datasets offers an alternative, upstream approach

In contrast to XAI tools and other technology-based solutions, the public release of datasets in science, medicine, and engineering offers an upstream solution. This human-centric approach, which focuses on better understanding biases in health data, maximizes the number of investigators involved and leverages their cognitive and social diversity to examine clinical data. There exist several examples throughout history whereby research necessitated multiple teams to examine the same dataset to ultimately reach an agreement. The case of right heart catheterization was notable: repeated analyses of the original dataset, collected by Connors et al. in 1996 [12], yielded conflicting results. This confusion left clinicians needing clarification about the

effect of the procedure for years. Untangling the confounding factors to reach our current understanding took several teams of biostatisticians. In sum, the coordinated efforts of many investigators, in an iterative learning process, are required to achieve consensus in data analysis and interpretation. In recent years [13,14], additional upstream approaches such as the embedded ethics methodology have been proposed to ensure that interdisciplinary ethical inquiry and deliberation are integrated into AI and healthcare technology development processes starting at project ideation [15]. Others have advanced approaches such as algorithmic impact analysis, which seeks to develop robust public interest methodologies to better understand the impact of AI and automated decision-making systems on people's lives and society at large [13].

## The current landscape of clinical data science research

In the past five years, several organizations–including the National Institutes of Health (NIH), European Commission, and World Economic Forum [16]–have recommended a shift to Open Data, a movement whose goal is to increase the release of FAIR (Findable, Accessible, Interoperable, and Reusable) [17] datasets in scientific research. In particular, large investments have been made in the biomedical sciences [18,19]. However, the clinical data science landscape remains highly siloed and opaque [20]. Expertise at the intersection of computer science and healthcare is currently concentrated among a few academic and industry research teams responsible for the preprocessing of data and the training of models [21,22]. Only researchers who are fortunate enough to be aware of a dataset's existence, to be granted access to it, and to have sufficient funding to afford the associated licensing fees and computing infrastructure, can effectively leverage it in practice. Therefore, in far too many instances, datasets are often inaccessible to investigators outside the very research team that curated them. For example, researchers who are not clinicians (e.g., computer scientists at MIT working on diabetic retinopathy in Uganda) often have limited access to primary data. Thus, they must rely on second-hand knowledge from clinical investigators at their institution or in their network (e.g., physicians at Harvard Medical School) but whose domain expertise may have been gained from datasets and practices originated in a different context. For example, datasets developed in North America may have significant limitations when used to train models to be implemented in East Africa and vice versa. Such secondary data analysis requires a deep understanding of the data curation process, including biases in data collection and artifacts in clinical measurements, which may vary locally by medical site or by region. Thus, knowledge transfer alone is insufficient to safeguard against the spread of biases. There are numerous reasons why bias occurs in clinical models. For instance, a decision-support model to prioritize screening for diabetic retinopathy that does not appropriately account for differences in the frequency of specialty visits among patients may result in selection bias. This example points to the need for familiarity with relevant socio-demographics and patient care-seeking behavior. Bias also can manifest when analysts inherit datasets without any background about the underlying environment. In such situations, they risk not only using a training dataset that is ill-suited for the target population, but also failing to understand the limitations of their model because local features have not been considered. Ideally, researchers will seek to interface with the team responsible for primary data collection. Without such a dialogue, external teams run the risk of unsafely deploying algorithms that do not generalize well out-of-distribution, for the cohorts of patients they care for.

## The promise of new NIH data-sharing policies

The recent imperative by the NIH to share scientific data publicly underscores a significant shift in academic research [23]. Effective as of January 2023, it emphasizes that transparency in

data collection and dedicated efforts towards data sharing–with other investigators and the broader public, including citizen scientists–are prerequisites for translational research, from the lab to the bedside. Certain fields of healthcare have paved the way: workshops on data ethics, privacy, consent, and anonymization have been organized in radiology [24]; a common ontology has been developed for data standardization in radiation oncology [25]; multi-center data-sharing platforms have been designed for collaboration in sleep medicine [26]; and distributed learning networks have recently been proposed as a solution to preserve the privacy of patients' EHR [27]. In the long run, requirements such as submitting a Data Management and Sharing Plan along with funding applications [28] will allow a more diverse population of researchers to interrogate both raw, unprocessed, and curated, pre-processed datasets. In light of the significance of data access to the mitigation of bias, we hypothesize that researchers who leverage existing open-access datasets rather than privately-owned ones are more diverse. We reason that the diversity of the backgrounds of open dataset users may in turn result into greater attention to equity in patient data collection and in a more compelling use of research data to address the pressing needs of the global population. This increased attention to issues of equity could also facilitate a more nuanced interpretation of study results and a stronger willingness to translate these results into practice.

### Structure of this paper

In this brief report, we propose to test this hypothesis in the transdisciplinary and expanding field of AI for Critical Care. Specifically, we compare the diversity among authors of publications leveraging open datasets, such as the commonly used MIMIC [29] and eICU databases [30], with that among authors of publications relying exclusively on private datasets, unavailable to other research investigators. To measure the extent of author diversity, we characterize gender balance, geographic diversity (i.e., the number of countries with which authors are affiliated and the income categories these countries map to), and the presence of researchers from minority serving institutions (MSI). Furthermore, we comment on the challenges of increasing the participation of researchers from underrepresented groups and suggest changes that can be made to the current Open Data strategy to enhance representation in authorship in the next decade. By evaluating the association between data accessibility and author diversity, our study pinpoints actionable steps that the broader field of clinical AI can take to foster inclusion in the scientific community and mitigate blind spots in data preparation and/or model development.

### Methods

Our scripts and datasets are publicly available on GitHub:
https://github.com/anpetushkov/OpenVsPrivateDatasets

### Data

We leveraged PubMed [31] to select research studies at the intersection of AI and Critical Care published between 2010 and 2022. We created two separate queries to derive (1) a list of publications related to AI and (2) a list of publications addressing topics in critical care medicine. We used the same process as Celi et al. [19] for the AI-specific query since the authors' model identified AI-related publications with suitable performance for our task (AUROC = 0.96). Query search terms specific to critical care were selected based on Van de Sande et al. [32] and vetted by two physician authors on our team, LAC and JG. Subsequently, we merged the two publication lists, thereby capturing only the studies related to both fields. Further, we split the resulting set of studies into two groups: "treatment" and

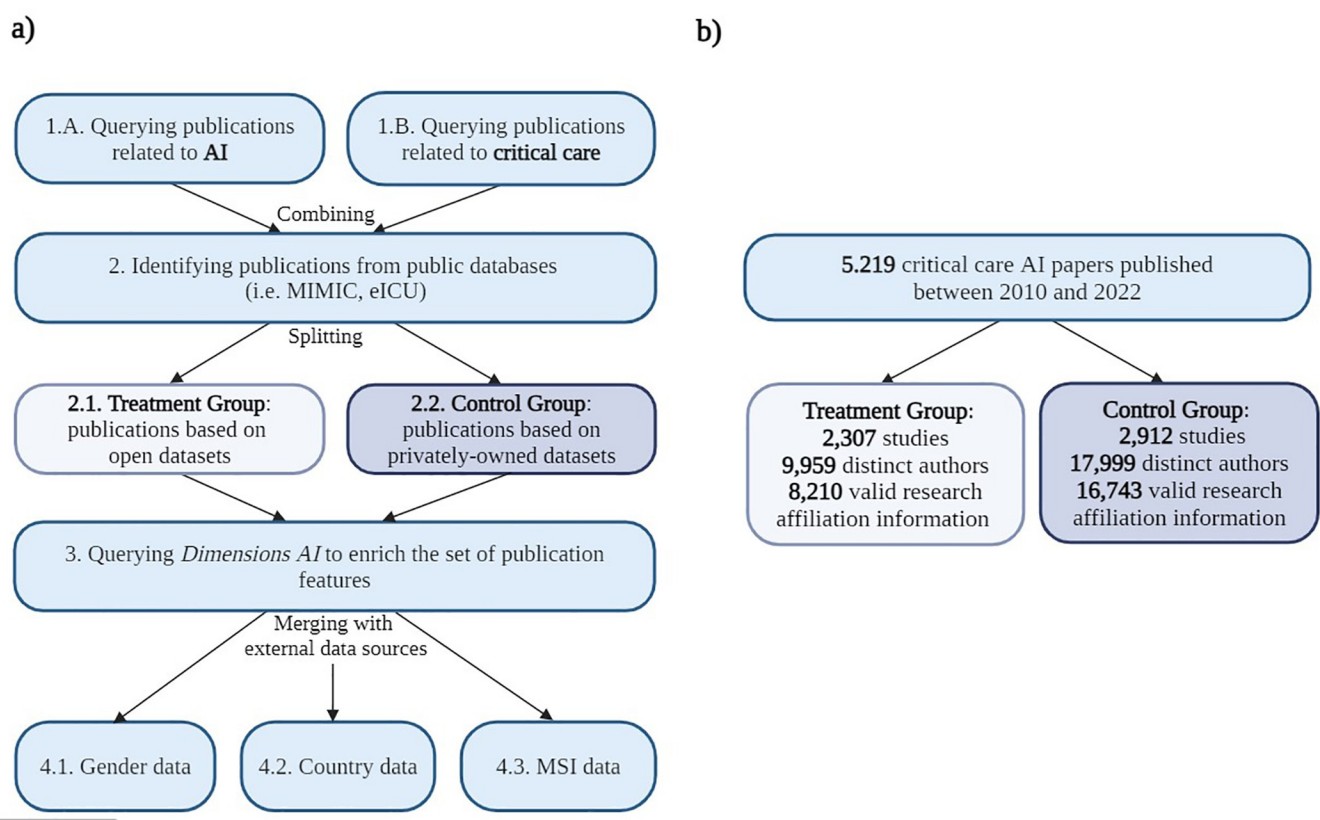

**Fig 1.** Flow diagram illustrating **(A)** the methodology for the analysis of authorship diversity in scientific publications at the intersection of AI and Critical care and **(B)** the number of publications considered.

"control." The treatment group comprised publications leveraging either of the two major critical care databases currently in open access, i.e., MIMIC and eICU. We used dataset-specific queries from Google Dataset Search [33] to derive a list of works leveraging MIMIC or eICU critical care databases. Conversely, the control group consisted of publications based on privately-owned datasets that are unavailable to researchers other than the primary investigators. To avoid leakage, we confirmed that the two groups were mutually exclusive, i.e., no publication belonged to both. For studies in the control group, we first downloaded their unique PMID identifiers from PubMed, owing to the large sample size. Then, we used the Dimensions AI platform, an interlinked research information system provided by Digital Science, to collect metadata pertaining to each research article [34]. For studies in the treatment group, we performed the query via Dimensions AI directly as the sample size was much smaller. The platform was accessed in June 2023. Finally, we manually filtered the initial set of papers in the treatment group to exclude outliers and include only relevant MIMIC and eICU manuscripts. Three team members (AP, CL, and GL) completed this manual validation task independently before reconvening and reaching a consensus. Details about the creation of the study dataset are available in **Fig 1A**. Of note, we did not curate the MIMIC and eICU databases ourselves but provide a detailed description of the relevant studies to guide the readers who want to learn more about data collection and preprocessing in **S1 Text**.

## Labeling of author gender

Each author's first name was processed by the Genderize.io Application Programming Interface (API) (**Fig 1A**). The API is based on a global collection of first names that have been manually annotated and linked to their most likely gender. Building on this international database of first names, the probability that an author is a woman or a man can be derived from the API. The API returns an "unknown" label when the uncertainty is too high. We assigned the most probable gender label associated with each author's first name ("female," "male," or "unknown"). Importantly, Genderize.io is considered a reliable software to infer gender based on names. We chose to rely on the Genderize package rather than on the Gender API or the gender R package because a recent evaluation study by VanHelene et al. [35], based on a sizable multi-national dataset of over 32,000 gender-labeled clinical trial authors, reported it was generally more accurate than its counterparts (overall accuracy of about 96.6%; Gender API: 96.1%; gender R: 85.7%), while also being less expensive to use.

## Labeling of minority-serving institutions

To measure the extent of the representation of minority-serving institutions (MSI) within the control and treatment groups, we developed our own fuzzy-matching pipeline between a pre-specified list of institutions and each author's affiliation(s). In particular, we built upon the fuzzy-match Python package [36], specifically the Levenshtein Partial Ratio Function with a matching threshold of 97 percent. The list of MSIs used in this study was obtained by combining two data sources: the 2020 list from [37] comprising 774 distinct MSIs and the 2022 list from [38] containing 865 distinct MSIs (**Fig 1A**). A total of 566 institutions were shared by the two sources (according to our exact and fuzzy matching process based on the institution's name), while the remaining MSIs were unique to each list. The integration of these two data sources allowed for a more comprehensive set of MSIs.

To confirm the accuracy of the mapping between institutional affiliations and their potential MSI status, we performed manual verification of outputs from the fuzzy matching process for both the 2020 and 2022 MSI datasets. In cases when an author's institutional affiliation was incorrectly mapped to an MSI, we rectified the mistake manually. Verification was limited to reducing false positives, i.e., we only determined institutional affiliations that were erroneously linked with MSIs. However, our matching process was deemed comprehensive, since we selected a high threshold value of 97 percent to limit the number of false negatives.

## Labeling of institutions based in low- and middle-income countries

The 2022 World Bank country classification was used to map countries associated with researcher affiliations to the low- and middle-income category (LMIC) or the high-income category (HIC) (**Fig 1A**). Countries are ranked according to their gross national income. For authors with multiple affiliations, each was considered separately and mapped to the corresponding income category.

## Diversity metrics

A total of three diversity metrics were considered. First, for each paper, we quantified the overall number of authors, the number of probable women among the authors, and whether the first/last author was likely a woman. For both the control and treatment groups, we derived the proportion of probable women by article and the overall percentage of articles featuring an author who was likely a woman as the first and/or last author. Second, for each paper, we measured the number of authors affiliated with an institution based in an LMIC and whether the

first/last author was based in an LMIC. For both groups, we derived the proportion of LMIC authors by article and the overall percentage of articles featuring an LMIC author in a leading role. Third, for each article, we quantified the number of authors affiliated with an MSI and the MSI status of the first and/or last author's affiliation. For both groups, we similarly derived the proportion of MSI authors by article and the overall percentage of articles featuring MSI authors. Note that papers whose first or last author had an unknown gender or unidentifiable LMIC or MSI status based on their affiliation were excluded from the corresponding analyses.

## Statistical analysis

For each of the three diversity metrics of interest (i.e., gender representation, geographic diversity, and MSI status), we performed a one-sided proportional Chi-squared test of independence to determine if there was a significant difference between the control and treatment groups. We provide further details about the assumptions underlying the statistical tests used in our study in S2 Text. We set the threshold for statistical significance to 0.05, following common practice. A p-value less than 0.05 would thus indicate a statistically significant difference between the control and treatment groups (e.g., in terms of the representation of women, LMIC, or MSI authors), in favor of the latter. We applied a Bonferroni correction to handle multiple hypothesis testing. We conducted three sensitivity analyses to assess the impact of missing data and to test the robustness of our results–in terms of both gender representation and geographic diversity. In the first (respectively, second) counterfactual scenario for gender representation, we assumed that all authors whose gender could not be inferred were women (respectively, men). In the third counterfactual scenario, we used mean imputation to derive gender labels, i.e., using the distribution of either the control or treatment group, whichever the article belonged to. Similarly, in the first counterfactual scenario for geographic diversity, we assumed that none of the papers with missing author affiliation had any authors from LMICs. Conversely, in the second counterfactual scenario, we assumed that all of these papers had at least one author affiliated with an institution based in an LMIC. Lastly, in the third counterfactual scenario, we assumed that missing income category labels could be imputed via dataset-specific distributions derived from labeled data, i.e., using either that of the control or treatment group, depending on the group the article belonged to (Fig 1B).

## Results

Overall, we identified 5,219 critical care AI papers, including 2,912 studies in the control group (i.e., 55.8%) and 2,307 studies in the treatment group (i.e., 44.2%). The control and treatment groups comprised 17,999 and 9,959 distinct authors, respectively; among them, 16,743 (93.0% of the control group) and 8,210 (82.4% of the treatment group) had available research affiliation information. A total of 562 authors appeared in both groups, representing 3.4% of the control group and 6.8% of the treatment group. In the treatment group, the three leading venues were all preprint servers, accounting for 29.3% of all articles: arXiv (20.6%), Research Square (6.4%), and medRxiv (2.3%). Following these, the next three most popular venues were journals, accounting for 4.4% of all articles: Frontiers in Medicine (1.8%), Scientific Reports (1.3%), and, notably, Critical Care Medicine (1.3%), the flagship journal in the field. Together, these six venues accounted for 778 (33.7%) of all articles in the treatment group, suggesting that authors leveraging open datasets publish their work in a great diversity of outlets. In contrast, in the control group, the top venue was the IEEE Engineering in Medicine and Biology Society Conference (EMBC, 3.4%), followed by two journals, PLOS One (2.4%) and Scientific Reports (2.3%). Collectively, the top three forums represented 238 (8.2%) of all articles in the control group. Beyond the journal Critical Care Medicine, which accounts

**Table 1. Characteristics of papers and authors featured in the treatment and control groups.**

**Table 1A. Characteristics of authors in the treatment and control groups.**

| Diversity metric and author role | | Treatment group MIMIC/eICU (n / non-missing) | Control group (n / non-missing) |
|---|---|---|---|
| Probable woman (inferred gender) | Any author | 27.8% (2438 / 8758) | 30.8% (4710 / 15285) |
| LMIC* | Any author | 6.7% (547 / 8210) | 3.5% (590 / 16743) |
| MSI* | Any author | 8.6% (190 / 2207) | 5.6% (277 / 4914) |

**Table 1B. Characteristics of papers in the treatment and control groups.**

| Diversity metric and author role | | Treatment group MIMIC/eICU (n / non-missing) | Control group (n / non-missing) |
|---|---|---|---|
| Probable woman (inferred gender) | Any Author | 71.8% (1024 / 1426) | 73.0% (1330 / 1823) |
| | First Author | 28.1% (401 / 1426) | 30.9% (564 / 1823) |
| | Last Author | 23.6% (336 / 1426) | 21.7% (395 / 1823) |
| LMIC* | Any Author | 10.1% (150/ 1487) | 6.2% (168 / 2694) |
| | First Author | 7.9% (117 / 1487) | 4.8% (130 /2694) |
| | Last Author | 7.8% (116 / 1487) | 4.6% (123 / 2694) |
| MSI* | Any Author | 27.6% (126 / 456) | 17.7% (175 / 991) |
| | First Author | 7.9% (36 / 456) | 4.7% (47 / 991) |
| | Last Author | 6.1% (28 / 456) | 5.5% (55 / 991) |
| LMIC* & woman (inferred gender) | Any Author | 6.2% (59 / 945) | 3.8% (64 / 1698) |
| MSI* & woman (inferred gender) | Any Author | 20.6% (35 / 170) | 34.5% (86 / 249) |

*LMIC: Low- and Middle-Income Country, *MSI: Minority Serving Institution.

*Note 1*: LMIC-related statistics were determined based on papers with non-missing affiliation and hence non-missing country information. Note that 368 (8%) of all papers had at least one author with missing country information. This missing data pattern affected 185 (11.1%) and 183 (6.4%) papers from the treatment and control groups, respectively.

*Note 2*: In the treatment group, 104 papers (7.0%) had only one author. In the control group, 178 papers (6.6%) had only one author.

for 68 articles (2.3%), other popular venues included the Journal of Clinical Monitoring and Computing (1.5%) and Computers in Biology and Medicine (1.2%). Overall, the six main venues accounted for 385 (13%) of all articles in the control group, underscoring the heterogeneity of outlets in which authors leveraging private datasets publish their work as well. The comprehensive breakdown of author characteristics for each group is detailed in **Table 1**, while the full distribution of conference venues and journals is available in our GitHub repository. For each group, the distribution of papers among the top 10 venues is available in **S1 Fig** (treatment) and **S2 Fig** (control).

## Gender

The proportion of papers with at least one author inferred to be a woman was qualitatively comparable between the two groups, albeit slightly higher in the control group (73.0% vs. 71.8%, z = -0.726, p = 0.468). The representation of women among the first and the last authors was similar between the two groups (28.1% vs. 30.9%, z = -1.74, p = 0.0811; 23.6% vs. 21.7%, z = 1.28, p = 0.199, respectively). Importantly, in both groups, the proportion of women serving as a last author (overall average of 22.5%), often reflecting a senior research leadership role, was lower than that of women serving as a first author (overall average of 29.7%), generally awarded to the person leading study design and analysis. This difference was more pronounced (9.2 vs. 4.5 percentage points) in the control than in the treatment group. The results

of our three sensitivity analyses–assuming in turn that authors whose gender could not be inferred by the Genderize package were women (S1 Table) or men (S2 Table) or imputing missing gender labels by sampling from labeled examples (S3 Table)–were similar to those obtained in the main analysis, restricted to authors whose gender could be inferred. Thus, we believe that our conclusions with respect to the gender dimension are robust to potential mis-classification error.

## Low- and middle-income countries (LMIC)

Overall, out of the 4,181 articles with non-missing affiliations included in our study, 318 (i.e., 7.6%) had at least one LMIC author. The proportion of authors from LMICs was substantially higher in the treatment than in the control group (10.1% vs. 6.2%, z = 4.5, p<0.001). Moreover, we found that the diversity of first and last authors in terms of country of affiliation was greater in the treatment group, i.e., among studies leveraging MIMIC and eICU open critical care datasets. Indeed, 7.9% (vs. 4.8%, z = 4.30, p<0.001) of papers in the treatment group had their first author affiliated with an LMIC country. Furthermore, 7.8% (vs. 4.5% z = 3.14, p<0.001) had their last author affiliated with an LMIC country. The first sensitivity analysis (S4 Table), imputing missing data using the distribution based on labeled samples, also led to the conclusion of a greater representation of LMIC authors among researchers leveraging open datasets (overall: 10.1% vs. 8.3%, z = 4.5, p<0.001). The second sensitivity analysis (S5 Table), which made the optimistic assumption that all papers with missing affiliation information had authors from LMICs, confirmed the robustness of our results (overall: 42.0% vs. 16.7%, z = 23.5, p<0.001). The third sensitivity analysis (S6 Table), which made the pessimistic assumption that none of the papers with missing country information had authors from LMICs, yielded results qualitatively similar to the main analysis, albeit not statistically significant (overall: 6.5% vs. 7.3%, z = 1.1, p = 0.14; first: 4.6% vs. 4.9%, z = 1.3, p = 0.10; last: 4.6% vs. 5.2%, z = 0.77, p = 0.22).

## Minority serving institutions (MSI)

Our analysis of authorship among MSIs was restricted to the United States (US). The control group comprised 4,914 distinct authors with institutional affiliations within the US for a total of 16,743 distinct authors with an affiliation worldwide (i.e., 29.3%). Among them, 277 different authors were affiliated with MSIs, accounting for approximately 5.6% of the total. In contrast, the treatment group comprised 2,207 authors with institutional affiliations within the US, for a total of 8,210 distinct authors with non-missing affiliations worldwide (i.e., 26.9%). Among them, 190 different authors were affiliated with MSIs, accounting for approximately 8.6% of the total. Thus, the proportion of US-based authors affiliated with an MSI was 1.5 times higher among articles in the treatment than in the control group; this difference was statistically significant, suggesting that open data resources attract a larger pool of participants from minority groups (z = 4.69, p<0.001).

  In addition to overall statistics, we characterized the involvement of MSI researchers at the team level, i.e., per paper. The control (resp. treatment) group consisted of 991 (resp. 456) distinct papers with at least one author having an institutional affiliation in the US, out of 2,877 (resp. 1,672) papers worldwide (i.e., 34.4% and 27.3%, respectively). Among these papers, 175 (resp. 126) had at least one author affiliated with an MSI, representing approximately 17.7% (resp. 27.6%) of the total US research output in critical care AI involving the use of private (resp. open) databases. Thus, the proportion of papers featuring US-based authors affiliated with an MSI was 1.6 times higher in the treatment than in the control group; this difference

was statistically significant, suggesting that MSI researchers effectively benefit from open data resources, which further translates into publications and preprints (z = 4.34, p<0.001).

Out of the 991 distinct papers in the control group, the percentage of papers with an MSI-affiliated first author reached only 4.7% (47 / 991). The proportion of MSI researchers serving as first authors was significantly greater (z = 2.40, p = 0.008) in the treatment group, reaching 7.9% (36 / 456). This result suggests that barriers remain for MSI-affiliated authors to lead research studies when the underlying datasets are inaccessible to the broader public. In contrast, opening critical care datasets can bolster the participation of MSI researchers as first authors. Of note, the percentage of papers with an MSI-affiliated last author was larger (z = 0.45, p = 0.327) in the treatment group (28 / 456, i.e., 6.1%) than in the control group (55 / 991, i.e., 5.5%), but this difference was statistically insignificant owing in part to a reduced sample size in the analysis focused on MSI representation.

### Intersectionality

**Gender and LMIC.** Of the 2,643 papers with complete author data regarding both gender and LMIC status, a clear difference emerged between the control and treatment groups, with respect to the intersectional representation of researchers. In the treatment group comprising 945 papers, 59 (i.e., 6.2%) featured at least one woman and at least one LMIC-based researcher among the authors. In contrast, in the control group comprising 1,698 papers, only 64 (i.e., 3.8%) included both a woman and an LMIC-based researcher. This difference was statistically significant (z = 2.78, p = 0.003), underscoring greater intersectional diversity among authors who leveraged the publicly available MIMIC and eICU databases than among authors in the control group, who relied exclusively on private datasets.

**Gender and MSI.** Among the 277 authors in the control group affiliated with an MSI, 249 authors had non-missing gender information and 86 (i.e., 35%) were women. In contrast, among the 190 authors in the treatment group, 170 authors had non-missing gender, and 35 were women (i.e., 21%). The difference was statistically insignificant (z = -3.09, p = 0.999); hence there was no evidence of greater intersectional diversity, by gender and MSI, in the treatment group. Nonetheless, the sample sizes resulting from multiple stratifications were quite small, therefore limiting statistical power. Thus, future efforts should focus on monitoring trends over time to gather more evidence about differences in the representation of authors, with a focus on the intersectionality of their identities.

### Discussion

Our comparative analysis revealed a greater contribution of authors from LMICs and MSIs among researchers leveraging open critical care datasets than among those relying exclusively on private data resources. The participation of women was similar between the two groups, albeit slightly larger in the treatment group. Notably, although over 70% of all articles included at least one author likely to be a woman, they served as a first or last author in less than 25% of those articles. Thus, our study highlights the value of the Open Data strategy for underrepresented groups, while also quantifying persisting gender gaps in academic and clinical research at the intersection of computer science and healthcare. In doing so, we hope our work points to the importance of extending open data practices in deliberate and systematic ways.

While incorporating AI into healthcare is a technically challenging endeavor, its success depends not only on the performance of clinical models but also on the humans interfacing with them. Clinical models are a reflection of the patient data they are trained upon. The people collecting, processing, and analyzing the data all play a role in rendering the final representation of patients underlying inference and prediction tasks. Therefore, cognitive diversity

among researchers responsible for study design and data examination will facilitate a more thoughtful investigation of potential pitfalls encoded within clinical data.

Critical care research is still highly imbalanced. For example, while the incidence and mortality of sepsis is the highest in sub-Saharan Africa and other low- and middle-income countries, over 75% of clinical studies underlying the 2021 sepsis guidelines were conducted in high-income countries [39]. As the complexity of critical care data has increased, so has the complexity of the biases introduced: because of pronounced imbalances in the patient populations featured in research datasets [40], their identification can be difficult. Our research shows that open access to critical care data can change the status quo. With the public release of datasets such as MIMIC and eICU, we found that participation from authors based in LMICs or affiliated with MSIs can be greatly improved.

Open data offers a resource for data scientists and healthcare specialists to develop skills that are essential to patient care in the digital health era. However, the transparent release of datasets on freely-accessible cloud platforms is not sufficient in itself to generate meaningful knowledge in the biomedical sciences. Beyond data sharing and collaboration across institutions, improvements in education and research should be sought at multiple stages, starting with outreach programs aimed at diversifying teams of clinicians and engineers as well as continued education to raise awareness about both persisting and evolving health disparities. For instance, the INFORMED fellowship in oncology data science offered by the National Cancer Institute (NCI) constitutes an excellent model to be replicated elsewhere [41,42]. To sustain LMIC participation in critical care research, reducing the barrier to learning, engaging with, and publishing in, the digital health field is vital. Models are often built in one site but deployed in others. Therefore, it is crucial to enable teams serving at medical centers with fewer resources to examine distributional shifts between their local data and the data originally used for training and validation and to evaluate model performance locally [43]. Furthermore, temporal evaluations of subpopulation shifts (i.e., related to variation in patient sociodemographics and/or clinical profiles) and calibration drifts (possibly related to the former or to changes in clinical practice, outcome detection tools etc.) must be continuously performed. These checks can help detect the emergence of new disparities and measure the effectiveness of interventions aiming to correct for those that were previously identified [44]. With the increasing digitization of medical records, the hope is that the community of health informatics researchers will broaden and help break institutional silos at each site. While continuing to advocate for the collection of comprehensive clinical datasets appropriately reflecting the target populations, investing in implementation science and striving to integrate clinical models into healthcare systems should also be prioritized.

Although the differences observed between works leveraging open vs. private datasets are striking, we acknowledge three key limitations that affect the precision of the prevalences reported in our study. First, the algorithm that classified an author's likely gender was trained using binary gender labels on a researcher's first name, which provides an imperfect proxy for a complex attribute such as gender identity. While common in bibliometric analyses, this method overlooks nuances in how individuals self-identify and excludes those who do not fit into binary gender categories. Going forward, integrating survey data from individual researchers could enable a more accurate categorization, especially as gender model performance varies across languages and cultures. Specifically, although the Genderize package was found to be more accurate than other software packages in a prior study [35], differences in its ability to infer an author's gender on the sole basis of their first name persist across countries of origin. For instance, the performance of Genderize was found to be generally lower for authors of certain Asian origins, namely South Korean, Chinese, Singaporean, and Taiwanese researchers (82% for this subgroup vs. 96.6% on average) [35]. Second, representation was

assessed only in terms of the three following dimensions: gender, country income level, and minority-serving status of the author's institutions. Future work should move beyond such unidimensional definitions of diversity to capture the intersectional relationships that shape experiences in academia and clinical research. Third, the definition of MSIs was based on a US-centric designation. Future research should seek to assess geographic diversity more comprehensively, extending the analysis to countries outside the US; a globally inclusive framework is needed to understand how researchers from under-resourced institutions worldwide engage with open data within and across nations. Moreover, in order to clearly distinguish the effects on diversity outcomes attributable to open data from other possible confounding factors, a causal inferential approach would be required (e.g., through proper covariate adjustment, propensity score matching or weighting, doubly-robust methods). Beyond modeling, such a task would require an extensive data collection effort. Thus, a dedicated follow-up study is warranted to assess the causal effect of using open-access databases on diversity among research authors. It would involve adjusting for confounding factors affecting both our exposure of interest (i.e., the choice of addressing a research question using an open-access vs. a private database) and the three diversity outcomes considered in our study (i.e., gender of the authors, MSI status of their institutions, and LMIC status of the corresponding countries). For instance, the amount of resources available at the institutions with which the authors are affiliated (e.g., computing infrastructure, budget for data acquisition) may affect their decision to leverage an open-access vs. private database in their study. In parallel, the diversity of researchers involved in a study may depend on the type of institutions and available resources as well as on team-level funding. Additionally, in the name of novelty and originality, certain journals may prioritize the publication of studies based on private databases that are either explored by a team for the first time or remain largely unexplored. These same journals may also require prohibitive publication fees, preventing certain scholars from submitting their work for consideration when a waiver is not offered. Furthermore, the research topic being investigated, prior collaboration history, outreach efforts by database curators, and inherent team recruitment biases might confound the relationships observed in our study. Therefore, future work should focus on disentangling the role of open-access databases from these factors and seek to identify the share of differences in diversity outcomes directly and indirectly attributable to the availability of data sources. Mapping the network of critical researchers and their scholarly interactions, based on co-authorships as well as participation in conferences and workshops, may help with this task and would constitute an interesting extension of the present study.

## Conclusion

Without a concerted commitment to diversifying authorship, clinical AI research risks being confined to a limited group of institutions and individuals. Such homogeneity may introduce and perpetuate biases within AI systems, potentially exacerbating health disparities and reinforcing existing inequities in healthcare delivery. In response, we must actively promote gender representation and include voices from institutions that serve underrepresented populations, thereby incorporating essential perspectives that address the multifaceted dimensions of inequality.

The rise of open data platforms adhering to FAIR (Findable, Accessible, Interoperable, and Reusable) principles brings new opportunities for investigators worldwide to participate in biomedical research and knowledge creation. Future policy interventions, including by institutions and editorial boards, should consider the complex associations among access to open data, bias in the development and use of clinical models, and diversity in research groups. Going forward, it will be key to monitor progress frequently, particularly with respect to the

intersectional representation of authors–not only by gender, geography, and MSI status, but across the multiple dimensions that constitute a scientist's identity. Despite recent advances, sustaining an open and inclusive clinical AI ecosystem will require the retention of diverse talent, across geographies and career stages, in part through the provision of dedicated training. Policy and technology innovations must continue lowering barriers that prevent the broader, collaborative engagement of researchers with clinical data resources.

## Supporting information

**S1 Fig. Distribution of the top 10 venues among papers in the treatment group.**
(DOCX)

**S2 Fig. Distribution of the top 10 venues among papers in the control group.**
(DOCX)

**S1 Table. Results of the sensitivity analysis performed under the assumption that all authors with missing gender labels are women.**
(DOCX)

**S2 Table. Results of the sensitivity analysis performed under the assumption that all authors with missing gender labels are men.**
(DOCX)

**S3 Table. Results of the sensitivity analysis using the method of imputing missing data from the distribution of authors with gender labels.**
(DOCX)

**S4 Table. Results of the sensitivity analysis using the method of imputing missing data from the distribution of authors with labels (LMIC vs. not LMIC).**
(DOCX)

**S5 Table. Results of the sensitivity analysis performed under the assumption that all missing authors are from a LMIC.**
(DOCX)

**S6 Table. Results of the sensitivity analysis performed under the assumption that all missing authors are not from a LMIC.**
(DOCX)

**S1 Text. Brief overview of the four open-access critical care databases considered in our study.**
(DOCX)

**S2 Text. Further details about assumptions underlying the statistical tests used in the study.**
(DOCX)

## Author Contributions

**Conceptualization:** Marie-Laure Charpignon, Leo Anthony Celi, Marisa Cobanaj, Rene Eber, Jack Gallifant, Chenyu Li, Anton Petushkov.

**Data curation:** Marie-Laure Charpignon, Marisa Cobanaj, Rene Eber, Jack Gallifant, Chenyu Li, Gurucharan Lingamallu, Anton Petushkov.

**Formal analysis:** Marie-Laure Charpignon, Marisa Cobanaj, Chenyu Li, Gurucharan Lingamallu, Anton Petushkov.

**Funding acquisition:** Marie-Laure Charpignon, Leo Anthony Celi, Jack Gallifant.

**Investigation:** Marie-Laure Charpignon, Marisa Cobanaj, Chenyu Li, Anton Petushkov.

**Methodology:** Marie-Laure Charpignon, Marisa Cobanaj, Rene Eber, Jack Gallifant, Chenyu Li, Gurucharan Lingamallu, Anton Petushkov.

**Project administration:** Marie-Laure Charpignon, Rene Eber, Chenyu Li.

**Resources:** Marie-Laure Charpignon, Leo Anthony Celi.

**Software:** Marie-Laure Charpignon, Rene Eber, Chenyu Li, Anton Petushkov.

**Supervision:** Marie-Laure Charpignon, Leo Anthony Celi.

**Validation:** Marie-Laure Charpignon, Marisa Cobanaj, Rene Eber, Jack Gallifant, Chenyu Li, Anton Petushkov.

**Visualization:** Marie-Laure Charpignon, Marisa Cobanaj, Chenyu Li.

**Writing – original draft:** Marie-Laure Charpignon, Chenyu Li.

**Writing – review & editing:** Marie-Laure Charpignon, Leo Anthony Celi, Marisa Cobanaj, Rene Eber, Amelia Fiske, Jack Gallifant, Gurucharan Lingamallu, Anton Petushkov, Robin Pierce.

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
