## [Decision Letter · Decision Letter 0]

23 Apr 2024

Diversity and inclusion: A hidden additional benefit of Open Data

PDIG-D-24-00105

Dear Ms. Charpignon,

We are pleased to inform you that your manuscript 'Diversity and inclusion: A hidden additional benefit of Open Data' has been provisionally accepted for publication in PLOS Digital Health.

Best regards,

Miguel Ángel Armengol de la Hoz, Ph.D.

Section Editor

PLOS Digital Health

Reviewer Comments (if any, and for reference):

Reviewer's Responses to Questions

**Comments to the Author**

1. Does this manuscript meet PLOS Digital Health’s publication criteria? Is the manuscript technically sound, and do the data support the conclusions? The manuscript must describe methodologically and ethically rigorous research with conclusions that are appropriately drawn based on the data presented.

Reviewer #1: Yes

Reviewer #2: Partly

Reviewer #3: Yes

2. Has the statistical analysis been performed appropriately and rigorously?

Reviewer #1: Yes

Reviewer #2: No

Reviewer #3: Yes

3. Have the authors made all data underlying the findings in their manuscript fully available (please refer to the Data Availability Statement at the start of the manuscript PDF file)?

Reviewer #1: Yes

Reviewer #2: Yes

Reviewer #3: Yes

4. Is the manuscript presented in an intelligible fashion and written in standard English?

Reviewer #1: Yes

Reviewer #2: Yes

Reviewer #3: Yes

5. Review Comments to the Author

Reviewer #1: The manuscript contains important information about one of the benefits of open data, the participation of women and LMICs authors in critical care publications.

Publications produced with open data were compared with those produced with data obtained from private institutions, which are not open.

It results interesting that the large proportion of female authors was the same in both groups, as well as the small proportion of women being first of last author.

Is there any comment about the characteristics of the patients included in the open databases, compared to those of the patients included in the databases of private institutions?

The authors mention that having more authors from LMICs in publications based on open databases reduces clinical research bias, but the type of patients that are served in institutions with open or private data could influence the development of models that would be only applicable to the type of patients that generated the data.

Reviewer #2: Major Revisions

Dataset and Publication Selection Criteria:

Problem: The methodology for curating datasets like MIMIC and eICU is not explained.

Improvement: Provide a detailed description of the data selection and curation process to enhance reproducibility and clarify the scope of analysis.

Validation and Sensitivity Analysis of Gender Identification:

Problem: Reliance on the Genderize.io API without validation could introduce bias, especially across different cultural contexts.

Improvement: Validate the Genderize.io outputs against known gender data subsets and include sensitivity analyses to estimate and discuss the impact of potential misclassification.

Statistical Methodology and Assumptions:

Problem: The manuscript does not confirm whether the assumptions for using Chi-squared tests were met.

Improvement: Add a subsection on statistical methodologies that rigorously tests for assumptions, applies corrections for multiple testing (e.g., Bonferroni, FDR), and justifies the choice of these methods.

Attribution of Diversity Increases to Open Data:

Problem: The manuscript ambiguously attributes increases in diversity to open data use without accounting for other potential influencing factors.

Improvement: Integrate multivariable regression analysis to clearly distinguish the effects attributable to open data from other possible confounding factors.

Minor Revisions

Handling of Missing Data:

Problem: It is unclear how the manuscript handles missing data regarding gender, institutional affiliations, and country classifications.

Improvement: Clearly define the strategies for managing missing data, including the use of any imputation techniques, and discuss their impact on the analysis.

Comparative Analysis Framework:

Problem: The comparison between the impacts of open and privately-held datasets on diversity is not thoroughly explored.

Improvement: Establish a detailed analytical framework for comparing open vs. private datasets, focusing on the roles and contributions of diverse researchers.

Discussion of Confounding Variables:

Problem: The discussion section overlooks potential variables that could influence diversity outcomes.

Improvement: Elaborate on how research topic nature, outreach efforts by dataset curators, and inherent recruitment biases might confound the observed relationships.

Reviewer #3: The paper effectively addresses a critical issue in contemporary academic research, exploring the impact of open-access datasets on the diversity of authors in the field of artificial intelligence for critical care. The study provides valuable insights into the influence of data accessibility on author demographics, particularly highlighting the increased participation of researchers from low- and middle-income countries (LMICs) and minority-serving institutions (MSIs) when utilizing open datasets. Moreover, the findings underscore the persistent gender gaps in academic and clinical research, emphasizing the need for continued efforts to promote inclusivity. Overall, this research contributes significantly to the ongoing discourse on data transparency and its implications for equitable representation in scientific endeavors. Therefore, I recommend accepting this paper for publication.

6. PLOS authors have the option to publish the peer review history of their article (what does this mean?). If published, this will include your full peer review and any attached files.

**Do you want your identity to be public for this peer review?** For information about this choice, including consent withdrawal, please see our Privacy Policy.

Reviewer #1: **Yes: **Cleva Villanueva

Reviewer #2: No

Reviewer #3: No
